# Low Concentration Response Hydrogen Sensors Based on Wheatstone Bridge

**DOI:** 10.3390/s19051096

**Published:** 2019-03-04

**Authors:** Hongchuan Jiang, Xiaoyu Tian, Xinwu Deng, Xiaohui Zhao, Luying Zhang, Wanli Zhang, Jianfeng Zhang, Yifan Huang

**Affiliations:** 1State Key Laboratory of Electronic Thin Films and Integrated Devices, University of Electronic Science and Technology of China, Chengdu 610054, China; uestctxy@163.com (X.T.); xwdeng@uestc.edu.cn (X.D.); xhzhao@uestc.edu.cn (X.Z.); luyingz@163.com (L.Z.); wlzhang@uestc.edu.cn (W.Z.); 2National Key Laboratory of Science and Technology on Vacuum Technology and Physics, Lanzhou Institute of Physics, Lanzhou 730000, China; zhangjianfeng510@spacechina.com (J.Z.); huangyifan@spacechina.com (Y.H.)

**Keywords:** hydrogen sensors, PdNi thin films, Wheatstone bridge, low concentration

## Abstract

The PdNi film hydrogen sensors with Wheatstone bridge structure were designed and fabricated with the micro-electro-mechanical system (MEMS) technology. The integrated sensors consisted of four PdNi alloy film resistors. The internal two were shielded with silicon nitride film and used as reference resistors, while the others were used for hydrogen sensing. The PdNi alloy films and SiN films were deposited by magnetron sputtering. The morphology and microstructure of the PdNi films were characterized with X-ray diffraction (XRD). For efficient data acquisition, the output signal was converted from resistance to voltage. Hydrogen (H_2_) sensing properties of PdNi film hydrogen sensors with Wheatstone bridge structure were investigated under different temperatures (30 °C, 50 °C and 70 °C) and H_2_ concentrations (from 10 ppm to 0.4%). The hydrogen sensor demonstrated distinct response at different hydrogen concentrations and high repeatability in cycle testing under 0.4% H_2_ concentration. Towards 10 ppm hydrogen, the PdNi film hydrogen sensor had evident and collectable output voltage of 600 μV.

## 1. Introduction

Hydrogen (H_2_) is one of the most potential and cleanest energy sources [1]. Considering its low minimum ignition energy and wide flammable range (4–75%), a responsive sensor is essential to detect hydrogen concentration [1,2,3]. Fast and accurate detection of hydrogen concentration is vital to prevent hydrogen leakage when using liquid hydrogen and other aerospace operations in space [2]. Based on different working principles, the hydrogen sensor can be divided into electrical, electrochemical, optical type, work function based, etc. Among those sensors, work function based sensors typically need to work at elevated temperature in order to maintain high sensitivity [2,3]. While resistance based sensors are extensively applied due to ambient working temperature and convenience for measurement [2,4,5,6].

In the past decades, palladium (Pd) and palladium alloy have been widely used as the sensitive materials for metallic resistor type hydrogen sensor due to its high solubility for hydrogen [2,4,7,8]. Although pure Pd could deliver a high output, fast response and superior selectivity of hydrogen, there are still some shortcomings, such as structural deformations and the consequent hysteretic resistance behavior [2,4]. In order to solve these problems, nanostructured materials and Pd alloy have been used as hydrogen sensitive materials [9,10,11,12,13,14,15]. Ozturk et al. reported that when exposed to 10% hydrogen, the 6 nm Pd films deposited on flexible substrates exhibited good reversibility and good response [3]. Lee et al. reported that hydrogen absorption and desorption did not affect the macroscopic structural deformation of PdNi. The sensitivity decreased linearly with increasing Ni content in PdNi film [14]. Hoffheins et al. reported a sensor consisting of four pure resistors with Wheatstone bridge structure; two of them were shielded with the borosilicate-based glass as hydrogen permeation barrier and served as reference resistors compensating for changes in the resistance of the palladium due to temperature variations. The thick film sensor could detect hydrogen concentration at temperatures from 0 to 200 °C. However, the low concentration limit for hydrogen sensing was no less than 0.5% [16]. Moreover, the fabrication process of the sensor was not applicable for mass production.

In this study, a PdNi film hydrogen sensor with Wheatstone Bridge structure was designed and fabricated with micro-electro-mechanical system (MEMS) technology. The integrated sensors consisted of four PdNi alloy film resistors: two of them were shielded with silicon nitride film and used as reference resistance, while the others were used for hydrogen sensing. The output resistance signal was converted to millivolt output voltage signal for easy data acquisition. In addition, the property of the sensor was ameliorated by annealing treatment. Additionally, the performances of the Wheatstone bridge thin film hydrogen sensor were characterized and discussed.

## 2. Materials and Methods

### 2.1. Design and Fabrication of the Sensor

The schematic of the hydrogen sensor is shown in Figure 1. The sample was designed within the size of 3 cm × 3.25 cm × 0.43 mm. Single-sided polished silicon wafers were selected as the substrates. PdNi alloy film was used as the hydrogen sensing resistor. The bond pads of the PdNi were covered with Au film for precise resistance measurement. Additioally, line width of the PdNi line film resistance was 100 μm. The resistance of all of the resistors was designed as 8.4 kΩ.

As shown in the Figure 2, a Wheatstone bridge type hydrogen sensor was made up of four PdNi film resistors. An external voltage source *U_in_* was applied to Wheatstone bridge type hydrogen sensor. The output voltage (*U_out_*) can be expressed by:(1)$$U_{out}=\left(\frac{R_{1}+\Delta R_{1}}{R_{1}+R_{2}+\Delta R_{1}}-\frac{R_{3}}{R_{3}+R_{4}+\Delta R_{4}}\right)U_{in}-U_{0}$$
when the resistance of four resistors is the same (*R* = *R*_1_ = *R*_2_ = *R*_3_ = *R*_4_), it can be simplified as:(2)$$U_{out}=\frac{\Delta R}{2R+\Delta R}U_{in}-U_{0}=\frac{\Delta R}{2R+\Delta R}U_{in}$$
where *U_in_* and *U*_0_ are the voltage of external voltage source and initial bias of the Wheatstone bridge, respectively. Thus, the output voltage *U_out_* is directly related with Δ*R*. The output resistance signal was converted to millivolt output voltage signal for easy data acquisition.

The fabrication process of the hydrogen sensor was as follows: prior to the film deposition, the substrate was ultrasonic-cleaned successively with acetone, alcohol, and deionized water for 10 min. After that, the substrate was blow-dried with nitrogen. Next, PRI-4000A photoresist was spin-coated onto the substrate and patterned with lithographic technology. Afterwards, the PdNi film with the thickness of 100 nm was deposited by direct current magnetron sputtering and patterned with lift-off technique. Then, the silicon nitride layer with the thickness of 160 nm was deposited by RF magnetron sputtering and patterned on the two reference resistors. At last, Au film with the thickness of 220 nm was deposited by DC magnetron sputtering and patterned on the bond pads of PdNi resistors. Sputtering parameters of all the deposition process are included in Table 1. The crystal structure of the PdNi films was examined with X-ray diffraction (XRD, D/MAX-rA diffractometer, Rigaku, Japan) with a scan range from 35° to 50°. The surface morphology of the PdNi film was examined by scanning electron microscopy (SEM, Inspect F50, FEI, United States of America).

After the fabrication, the sample was vacuum annealed in nitrogen for 2 h at 300 °C. The sample was sliced into a single sensor with a size of 3 cm × 3.25 cm and gold wire was ball welded on the bond pad for data collection. The four resistors were connected accordingly with the external circuit to form the Wheatstone bridge. The photo of the fabricated hydrogen sensor is shown in Figure 3.

### 2.2. Hydrogen Sensing Tests

The hydrogen measuring system consisted of mass flow controllers (MFC), gas mix chamber, gas test chamber, temperature-controlled chamber, Keithley 2182 Nanovoltmeter and the power source (GPC-60300, Gwinstek, China), as shown in Figure 4. Before the measurement, the gas chamber was continuously purged with pure nitrogen for 2 h with the flow rate of 100 sccm (standard cubic centimeters per minute). Next, nitrogen and hydrogen gas with different ratios was transmitted to the mix chamber. After mixing, the gas mixture was delivered to the gas chamber. The flow rate the gas mixture was controlled to be 100 sccm. The output voltage of the Wheatstone Bridge was acquired with a LabVIEW program (National Instruments) under constant voltage mode with a source voltage of 10 V.

## 3. Results and Discussions

XRD pattern of the PdNi thin film was shown in Figure 5, two diffraction peaks localized at 40.64° and 47.69° reflect face-centered cubic (fcc) crystal structures of Pd and can be indexed as (111) and (200) planes, respectively. The peak positions are gradually shifted to higher angle with increasing Ni concentration in the PdNi films [11]. The full width half maximum (FWHM) of the peaks decreased after annealing, indicating the increment of the crystal size calculated with the Scherrer Equation [7].

The repeatability of the hydrogen sensor towards 0.4% hydrogen at different temperatures (30 °C, 50 °C and 70 °C) was revealed in the Figure 6. As the test temperature increased, the stability and zero drift of the sensor were partially improved despite the attenuation of the output signal. This phenomenon can be explained with the Sievert’ law and Sievert constant (K) which is defined as the solubility of gas molecules in metal materials:(3)$$\ln\;K=-\frac{\Delta_{s}H}{RT}+\frac{\Delta_{s}S}{R}$$
where Δ*_s_H* is the molar enthalpy of solution, Δ*_s_S* the molar entropy of solution, *R* the gas constant and *T* the environment temperature [3,17,18]. Equation (3) explains clearly that there was an inverse relationship between the temperature and sensitivity of the metal film resistance sensor [3]. The solubility of H in PdNi alloy films decreased with increased test temperature, which results in reducing maximum response of Wheatstone bridge type hydrogen sensor.

The response time and recovery time were defined as the consumed time to reach 90% of the final stable value [12]. When exposed to hydrogen, the output voltage of the sensor increased rapidly. After switching to pure nitrogen, the output voltage of the sensor decreased sharply, as shown in Figure 6 and Figure 7a. When the hydrogen concentration increased from 200 ppm to 800 ppm, the output voltage of (2) the unannealed hydrogen sensor increased from 10.225 mV to 15.434 mV; the response time and recovery time also decreased gradually, as shown in Figure 7a. These results indicate that with increasing hydrogen concentration, the diffusion rate of hydrogen and the rate of hydride formation increased [2,4,5,17]. With a same test time, the output value of unannealed sensor did not attain the equilibrium at 200 ppm. Compared with unannealed sample, the output response of the annealed sensor was somewhat reduced; however, its zero drift and repeatability were greatly improved. It has been reported that annealing treatment could alter the microstructure of Pd alloy film into influence H absorption characteristics. Hydrogen solubilities in dilute phase decrease with enhancing alloy homogeneity [19]. After annealing at 300 °C, the PdNi peak is apparently stronger due to the grain size growth in the Figure 5a. Due to the low annealing temperature, the microstructure of the film in the SEM image might not be significantly different. Grain size growth might be one of the factors causing change of H absorption. In addition, the internal stress in alloy film was released after annealing treatment, which might be another factor for the fast response time [8,20]. The output signal, the response and recovery time of the annealed hydrogen sensor all decrease after annealing treatment.

Figure 8 displays the response of the annealed hydrogen sensor at low H_2_ concentration. The annealed sensor still has a detectable output voltage of 600 μV at 10 ppm hydrogen. In addition, compared with single PdNi film resistance hydrogen sensor [12], the response time and recovery time of the Wheatstone bridge sensor were significantly reduced and the zero drift was greatly ameliorated. It should be noted that in sharp contrast to previous report [14,15,16], this sensor still had an obvious output response under 10 ppm hydrogen, which makes it applicable for wide range hydrogen concentration detection (10 ppm-0.4%). Furthermore, the sensor was fabricated on silicon substrates with MEMS technology, which makes it much easier for mass production and integration [21].

## 4. Conclusions

The highly sensitive Wheatstone bridge type hydrogen sensor based on PdNi alloy thin films was fabricated and tested. The output resistance signal was converted to millivolt output voltage signal for easy data acquisition. Compared with unannealed sample, the annealed sensor had excellent output response, faster response time and recovery time. Furthermore, the sensor demonstrated obvious output response with excellent stability and reliability over wide range of hydrogen concentration (10 ppm-0.4%). This sensor was fabricated on silicon substrates with MEMS technology, which makes it much easier for mass production and integration.

## Figures and Tables

**Figure 1 sensors-19-01096-f001:**
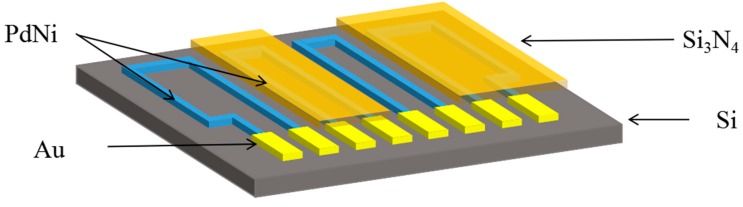
Schematic of the hydrogen sensors.

**Figure 2 sensors-19-01096-f002:**
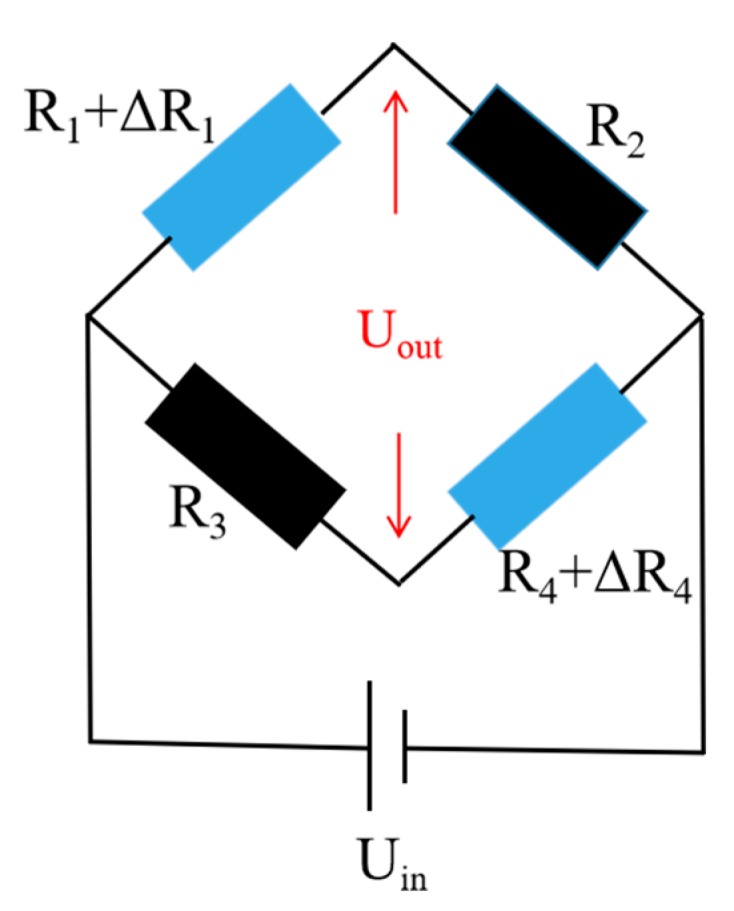
Structure diagram of Wheatstone bridge type hydrogen sensor.

**Figure 3 sensors-19-01096-f003:**
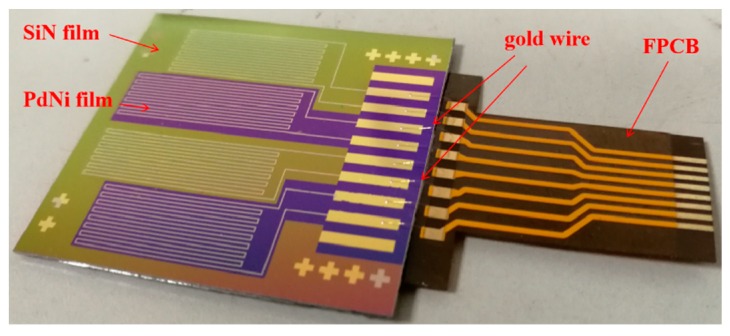
Photo of the fabricated hydrogen sensor.

**Figure 4 sensors-19-01096-f004:**
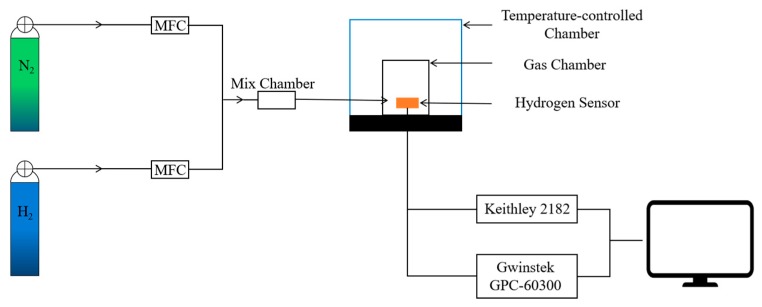
Schematic of the hydrogen measuring system.

**Figure 5 sensors-19-01096-f005:**
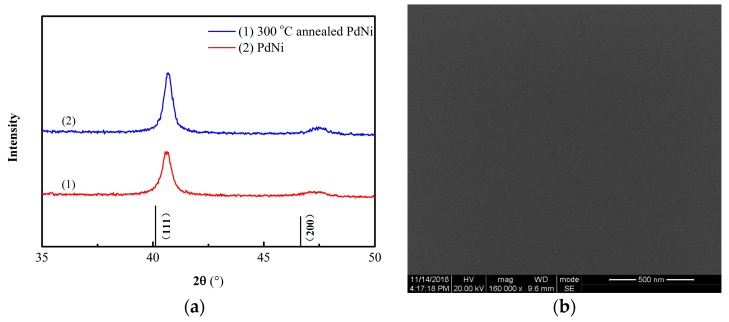
(**a**) X-ray diffraction (XRD) patterns of (1) PdNi and (2) 300 °C annealed PdNi film; (**b**) SEM image of PdNi thin film.

**Figure 6 sensors-19-01096-f006:**
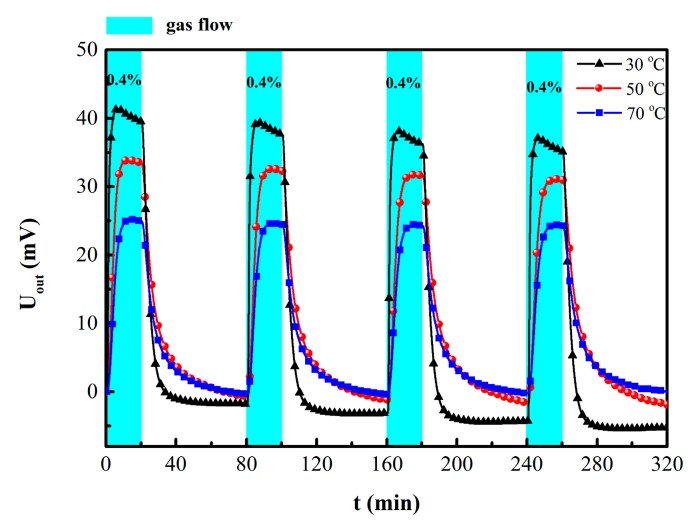
Repeatability of the hydrogen sensor at 30 °C, 50 °C and 70 °C under 0.4% H_2_.

**Figure 7 sensors-19-01096-f007:**
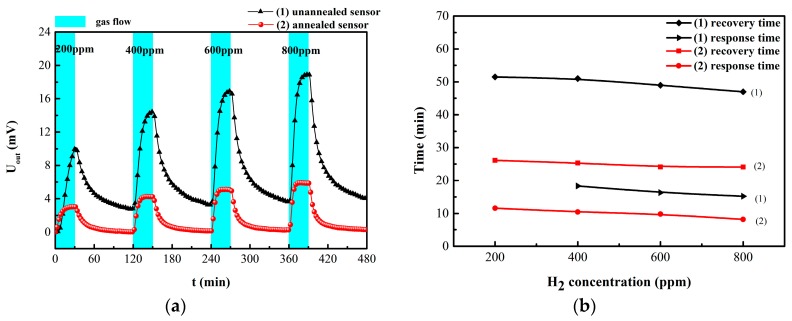
(**a**) Response curve of (1) unannealed sensor and (2) annealed hydrogen sensor at 50 °C under different H_2_ concentrations; (**b**) the response time and recovery time of (1) and (2).

**Figure 8 sensors-19-01096-f008:**
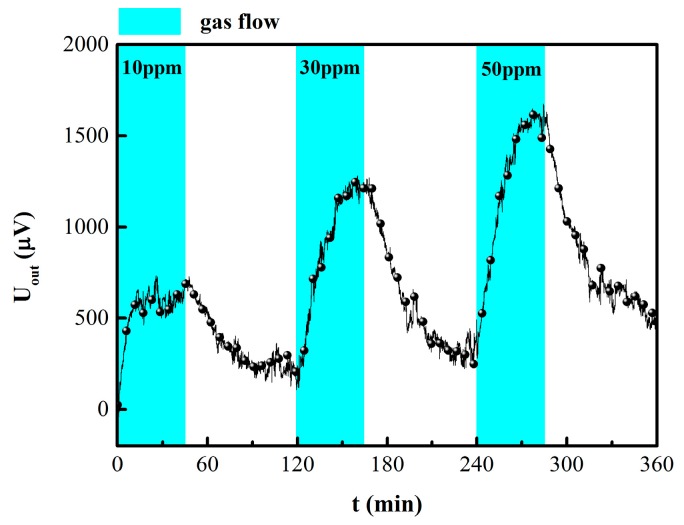
Response curve of the annealed hydrogen sensor at 50 °C.

**Table 1 sensors-19-01096-t001:** Sputtering parameters of different films.

Material	Base Pressure (Pa)	Sputtering Pressure (Pa)	Sputtering Power (W)	Temperature (°C)
PdNi	8 × 10^−4^	0.3	60	RT
Si_3_N_4_	8 × 10^−4^	0.5	200	RT
Au	8 × 10^−4^	0.3	60	RT

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
