# Peer review of "Low Concentration Response Hydrogen Sensors Based on Wheatstone Bridge"

_sensors, 2019, doi:10.3390/s19051096_

Reviewer 1 Report

Although, detection of H2 at low operating temperature is of real interest, the present manuscript suffer of several issues:

- beside XRD additional morphological investigations are needed in order to gain insights about the material morphological properties,

- the unannealed sensor the sensor response time cannot be evaluated at 200 ppm since equilibrium is not attained,

- what about water vapor influence since the authors talk about mass product and integration?

Author Response

Thanks for your hard work and useful comments on our paper. We have suitably addressed the reviewer’s comments and carefully revised the whole manuscript. Each change made as raised in the comments was outlined point by point below. And all of the revision have been highlighted in the revised manuscript.

Q1: Beside XRD additional morphological investigations are needed in order to gain insights about the material morphological properties.

A1: Thanks for your suggestion. Considering the Reviewer’s suggestion, PdNi film was studied by Scanning Electron Microscopy (SEM) as shown in Fig 5(b). The full width at half maximum (FWHM) of the peak after annealing was lowered and the crystal size calculated by the Scherrer equation did increase. Due to the low annealing temperature, there might be no significant difference in SEM images. We have cited other literature in the paper to discuss and analyze the effect of annealing on the sensor.

Fig 5 (b) SEM images of PdNi thin film

Q2: The unannealed sensor the sensor response time cannot be evaluated at 200 ppm since equilibrium is not attained.

A2: Thanks for your suggestion. It is really true as Reviewer pointed out that the response time of the unannealed sensor the sensor cannot be evaluated at 200 ppm because of not attaining equilibrium. In order to setting same interval time for comparison, it resulted in insufficient passing hydrogen time at 200 ppm and unstable maximum. We have deleted that response time and that value did not affect the subsequent test results. We can infer that the actual response time should be larger than the previous response time and longer than the response time of the annealed sensor. Moreover, we will completely test the performance of the unannealed sample at low concentrations in future research work.

Q3: What about water vapor influence since the authors talk about mass product and integration?

A3: Thanks for your suggestion. Compared with other application environments, the sensor would be used in spacecraft or space and water vapor content is less. According to other literature, humidity little effects the sensor. Even water vapor would increase the output response of the oxide semiconductor type sensor. It could be believed that water vapor on the sensor may have little effect. Furthermore, we will examine the impact of water vapor on the performance of the sensor in future research work.

Special thanks to you for your good comments.

We hope that the above careful modifications meet your requirements. Your comments are significant to improve our research in the future.

We look forward to receiving your good news soon.

Yours Sincerely

Hongchuan Jiang *, Xiaoyu Tian

Reviewer 2 Report

The manuscript is of interest for the journal but needs major revision prior to its being considered for acceptance. Below are the main comments that can help improve it.

1)  Page 1, abstract: “The interval two of them”. Please make sure if it’s “interval” (or “internal”?)

2) Page 1, introduction, line 6 from bottom: “ among those hydrogen sensor” (those sensors? Not sensor?)

3) Page 1, line 5 from bottom: “sensors typically needs to work” (need to work?).

4) Page 1, line 4 from bottom: “sensors are extensively applied due to its simple structure” (sensors… do to their simple structure?)

5) Reference list: the list of references is not properly organized! The authors must refer to the current issue of this journal in order to see how their references must be presented.

6) page 2, lines 5, 7 from top:  the authors must be aware of how to present the names of the others when those are cited. For example:  Ozturk et al, Lee et al, or Ozturk and coauthors… No first names or initials must be used in the text. Please check the entire text and make proper corrections.

7) page 2: “two of which was shielded” (two of which were?)

8) Introduction, last paragraph: the authors should add, in a clear way, what original and novel this manuscript demonstrates and reports for the reader. 1-2 short sentences added here would be enough, which clearly state the novelty and originality of this work (in comparison with the previous work)

9) Fig.2, caption: this reviewer sees no working principle of the sensor in Fig.2, while the figure apparently presents the structure of the sensor.

10) Table 1: “different functional film.” (different films?)

11) page 3: “four resistors was connected” (were connected?)

12) page 4, lines 5,6 from bottom: “peaks …. reflects” (peaks reflect?)

13) page 5, paragraph above Fig.6: “enhancing test temperature”. What is the meaning of “enhanced temperature”? This reviewer cannot understand why “increased” means “enhanced” in this manuscript (regarding temperature)

14) Fig.7 ,caption: the authors used incorrect chemical formula for hydrogen

15) page 6, lines 2-3 from top: “causing leading to change of H absorption” (English must be improved)

16) page 6: “wheatstone bridge sensor”. There is inconsistency with the other pages where “Wheatstone” is spelled with upper-case letter “W”

17) page 7, conclusions, line 2 from bottom: “Furthermore, The sensor” (why “The” and not “the”?)

Author Response

Thanks for your hard work and useful comments on our paper. We have suitably addressed the reviewer’s comments and carefully revised the whole manuscript. Each change made as raised in the comments was outlined point by point below. And all of the revision have been highlighted in the revised manuscript.

Q1: Page 1, abstract: “The interval two of them”. Please make sure if it’s “interval” (or “internal”?).

A1: Thanks for your suggestion. We are very sorry for misnomer and have exchanged “interval” to “internal”.

Q2: Page 1, introduction, line 6 from bottom: “ among those hydrogen sensor” (those sensors? Not sensor?).

A2: Thanks for your suggestion. We are very sorry for our incorrect writing and have exchanged “those hydrogen sensor” to “those sensors”.

Q3: Page 1, line 5 from bottom: “sensors typically needs to work” (need to work?).

A3: Thanks for your suggestion. We are very sorry for our incorrect writing and have exchanged “needs” to “need”.

Q4: Page 1, line 4 from bottom: “sensors are extensively applied due to its simple structure” (sensors… do to their simple structure?).

A4: Thanks for your suggestion. We are very sorry for the error in the expression.  Considering the Reviewer’s suggestion, we have deleted the phrase “its simple structure”. The resistance based sensors are extensively applied due to ambient working temperature and convenience for measurement.

Q5: Reference list: the list of references is not properly organized! The authors must refer to the current issue of this journal in order to see how their references must be presented.

A5: Thanks for your suggestion. We are very sorry for our incorrect writing and we have tried our best to correct usage mistakes in the manuscript.

Q6: page 2, lines 5, 7 from top: the authors must be aware of how to present the names of the others when those are cited. For example:  Ozturk et al, Lee et al, or Ozturk and coauthors… No first names or initials must be used in the text. Please check the entire text and make proper corrections.

A6: Thanks for your suggestion. We have made correction according to the Reviewer’s comments. We have exchanged “Sadullah Oztürk”, “Lee E” and “B.S. Hoffheins” to “Sadullah ”, “Lee ” and “ Hoffheins”. And we have checked the entire text and make proper corrections.

Q7: page 2: “two of which was shielded” (two of which were?).

A7: Thanks for your suggestion. We are very sorry for our incorrect writing and have exchanged “was” to “were”.

Q8: Introduction, last paragraph: the authors should add, in a clear way, what original and novel this manuscript demonstrates and reports for the reader. 1-2 short sentences added here would be enough, which clearly state the novelty and originality of this work (in comparison with the previous work).

A8: Thanks for your suggestion. We have re-written this part according to the Reviewer’s suggestion. We add the sentence “The output resistance signal was converted to millivolt output voltage signal for easy data acquisition. In addition, the performance of the sensor was ameliorate by annealing treatment.”.

Q9: Fig.2, caption: this reviewer sees no working principle of the sensor in Fig.2, while the figure apparently presents the structure of the sensor.

A9: Thanks for your suggestion. We are very sorry for misnomer and have exchanged “Working principle diagram” to “Structure diagram”.

Q10: Table 1: “different functional film.” (different films?).

A10: Thanks for your suggestion. We are very sorry for misnomer and have deleted the word “functional”.

Q11: page 3: “four resistors was connected” (were connected?).

A11: Thanks for your suggestion. We are very sorry for our incorrect writing and have exchanged “was” to “were”.

Q12: page 4: lines 5,6 from bottom: “peaks …. reflects” (peaks reflect?).

A12: Thanks for your suggestion. We are very sorry for our incorrect writing and we have exchanged “reflects” to “reflect”.

Q13: Page 5: paragraph above Fig.6: “enhancing test temperature”. What is the meaning of “enhanced temperature”? This reviewer cannot understand why “increased” means “enhanced” in this manuscript (regarding temperature).

A13: Thanks for your suggestion. We are very sorry for misnomer and have exchanged “enhanced” to “increased”.

Q14: Fig.7: caption: the authors used incorrect chemical formula for hydrogen.

A14: Thanks for your suggestion. We are very sorry for our incorrect writing and we have exchanged “H2” to “H2”.

Q15: page 6:lines 2-3 from top: “causing leading to change of H absorption” (English must be improved)

A15: Thanks for your suggestion. We are very sorry for misnomer and have deleted the phrase “leading to”.

Q16: page 6: “wheatstone bridge sensor”. There is inconsistency with the other pages where “Wheatstone” is spelled with upper-case letter “W”

A16: Thanks for your suggestion. We are very sorry for our incorrect writing and have exchanged “wheatstone” to “Wheatstone”.

Q17: page 7: conclusions, line 2 from bottom: “Furthermore, The sensor” (why “The” and not “the”?)

A17: Thanks for your suggestion. We are very sorry for our incorrect writing and have exchanged “The” to “the”.

We have tried our best to correct grammar errors and usage mistakes in the manuscript. Special thanks to you for your good comments.

We hope that the above careful modifications meet your requirements. Your comments are significant to improve our research in the future.

We look forward to receiving your good news soon.

Yours Sincerely

Hongchuan Jiang *, Xiaoyu Tian

Round  2

Reviewer 1 Report

The authors have improved the manuscript according with my suggestions.

Author Response

Dear Reviewer,

Thank you so much for your reviewing! We deeply appreciate your recognition of our research work. We think the reviewer’s comments were insightful and all the recommendations were taken into account in this revised manuscript. We appreciate their efforts in improving the quality of this article.

Yours Sincerely

Hongchuan Jiang *, Xiaoyu Tian

Reviewer 2 Report

The manuscript needs revision:

Lines 38-39:     While resistance based sensors are extensively applied due to ambient working temperature and convenience for measurement [2,3,5].

Line 49: "desorption didn’t affect" (did not! as this is a scientific text rather than colloquial language)

Line 63: "was ameliorate by" (was ameliorated by?)

Line 96: "of different film" (of different films?)

Lines 144-146: In order to setting same interval time for comparison, it resulted in insufficient passing hydrogen time at 200 ppm and unstable maximum.     (this sentence is unreadable at all and needs rewriting)

Author Response

Thanks for your hard work and useful comments on our paper. We have suitably addressed the reviewer’s comments and carefully revised the whole manuscript. Each change made as raised in the comments was outlined point by point below. And all of the revision have been highlighted in the revised manuscript.

Q1: Lines 38-39: While resistance based sensors are extensively applied due to ambient working temperature and convenience for measurement [2,3,5].

A1: Thanks for your suggestion. We have rewritten this part according to the Reviewer’s suggestion. We have exchanged the sentence to “Due to its ambient working temperature and convenience for measurement, resistance based sensors are extensively applied [2,3,5].”.

Q2: Line 49: "desorption didn’t affect" (did not! as this is a scientific text rather than colloquial language).

A2: Thanks for your suggestion. We are very sorry for our incorrect writing and have exchanged “didn’t” to “did not”.

Q3: Line 63: "was ameliorate by" (was ameliorated by?).

A3: Thanks for your suggestion. We are very sorry for our incorrect writing and have exchanged “ ameliorate” to “ ameliorated”.

Q4: Line 96: "of different film" (of different films?).

A4: Thanks for your suggestion. We are very sorry for our incorrect writing and have exchanged “ film” to “ films”.

Q5: Lines 144-146: In order to setting same interval time for comparison, it resulted in insufficient passing hydrogen time at 200 ppm and unstable maximum.     (this sentence is unreadable at all and needs rewriting).

A5: Thanks for your suggestion. We are very sorry for the error in the expression.  Considering the Reviewer’s suggestion, we have rewritten and exchanged the sentence to “In order to compare the two sensors under same interval test time, the unannealed sensor at 200 ppm did not reach a stable maximum response.”.

We deeply appreciate your recognition of our research work. We think the reviewer’s comments were insightful and all the recommendations were taken into account in this revised manuscript. We appreciate their efforts in improving the quality of this article.

We hope that the above careful modifications meet your requirements. Your comments are significant to improve our research in the future.

Yours Sincerely

Hongchuan Jiang *, Xiaoyu Tian

Round  3

Reviewer 2 Report

 The manuscript needs revision. There are still errors in the manuscript, in particular in the newly added sentences:

1) Due to its ambient working temperature and convenience for measurement, resistance based sensors are extensively applied

2) In order to compare the two sensors under same interval test time, the unannealed sensor at 200 ppm did not reach a stable maximum response.

These sentences either have grammatical errors (such as, e.g.: ... its ... sensors are ...) or are completely unreadable (sentence 2).

Author Response

Thanks for your hard work and useful comments on our paper. We have suitably addressed the reviewer’s comments and carefully revised the whole manuscript. Each change made as raised in the comments was outlined point by point below. And all of the revision have been highlighted in the revised manuscript.

Q1: The manuscript needs revision. There are still errors in the manuscript, in particular in the newly added sentences: 1) Due to its ambient working temperature and convenience for measurement, resistance based sensors are extensively applied.

A1: Thanks for your suggestion. We are very sorry for the error in the expression. Considering the Reviewer’s suggestion, we have rewritten and exchanged the sentence to “While resistance based sensors are extensively applied due to ambient working temperature and convenience for measurement [2,3,5].”.

Q2: 2) In order to compare the two sensors under same interval test time, the unannealed sensor at 200 ppm did not reach a stable maximum response.

A2: Thanks for your suggestion. We have rewritten this part according to the Reviewer’s suggestion. We have exchanged the sentence to “With same test time, the output value of unannealed sensor did not attain the equilibrium at 200 ppm.”

We deeply appreciate your recognition of our research work. We think the reviewer’s comments were insightful and all the recommendations were taken into account in this revised manuscript. We appreciate their efforts in improving the quality of this article.

We hope that the above careful modifications meet your requirements. Your comments are significant to improve our research in the future.

Yours Sincerely Hongchuan Jiang *, Xiaoyu Tian